# Interventions for depression and anxiety among people with diabetes mellitus: Review of systematic reviews

**Reginaldo Tavares Franquez[1], Isabela Muniz de Souza[1,2], Cristiane de Cássia Bergamaschi** **[1] \***

**1** Pharmaceutical Sciences Graduate Course, University of Sorocaba, Sorocaba, São Paulo, Brazil, **2** Dental School, University of Sorocaba, Sorocaba, São Paulo, Brazil

\* cristiane.motta@prof.uniso.br

**Data Availability Statement:** All relevant data are within the paper and its Supporting Information files.

## Abstract

This review of systematic reviews of randomized clinical trials summarized the available evidence regarding the effectiveness and safety of interventions to treat depression and/or anxiety in people with type 1 and type 2 diabetes. The sources of information searched were the Cochrane Library, MEDLINE, EMBASE, Web of Science and LILACS, until up to December 1st, 2022. The interventions were compared with placebo, active control or usual care. The measured primary outcomes were improvement in depression and anxiety remission, reduction of diabetes-specific emotional distress; and improvement in quality of life. Two reviewers, independently, selected the reviews, extracted their data, and assessed their methodological quality using AMSTAR-2. A narrative synthesis of the findings was performed, according to the type of intervention and type of diabetes. Thirteen systematic reviews that included 28,307 participants were analyzed. The reviews had at least one critical methodological flaw. Cognitive Behavioral Therapy improved the mainly depression, glycemic values (n = 5 reviews) and anxiety (n = 1), in adults and elderly with diabetes. Collaborative care (n = 2) and health education (n = 1) improved depression and glycemic values, in adults with diabetes. Pharmacological treatment (n = 2) improved depression outcomes only. The quality of the evidence was low to moderate, when reported. The interventions reported in literature and mainly the Cognitive Behavioral Therapy can be effective to treat people with diabetes and depression; however, some findings must be confirmed. This study can guide patients, their caregivers and health professionals in making decisions concerning the use of these interventions in the mental healthcare of people with diabetes.

**Protocol Registration:** PROSPERO (CRD42021224587).

## Introduction

Diabetes mellitus is a disabling long-term health condition that is common and growing. Globally, it affects 10.5% of the population and is the leading cause of lost disability-adjusted life years [1,2]. It was considered the ninth leading cause of death in 2019, with an estimated 1.5 million deaths caused by diabetes [3]. It is estimated that 536.6 million individuals

**Funding:** This study received a scholarship funded by the Governmental Program Graduate Education Institutions—PROSUC—CAPES/UNISO.

**Competing interests:** The authors have declared that no competing interests exist.

worldwide have diabetes, and it is expected that the number will increase to 783.2 million, in 2045 [2].

Psychological disorders can occur in people with diabetes of all ages [4]. Type 1 diabetes seems to be associated with a higher prevalence of psychological disorders than type 2 diabetes [5].

Type 1 diabetes is the most common endocrine disorder in children and the self-management of this condition can be difficult mainly in children and adolescents with psychological disorders. Adolescents are 2.3 times more likely to have psychological disorders than adults [6]. Depression and anxiety in children and adolescents with type 1 diabetes also had a negative impact on the management of this condition and on self-care behavior's [7,8]. Systematic review identified that adults with diabetes were approximately 1.5 times more likely to have anxiety symptoms [9]. Another systematic review demonstrated that people with anxiety disorders were found to have a 19% higher risk of diabetes [10].

Depression is a common co-morbidity in people with diabetes. Both diseases are growing rapidly and have a negative impact on the physical, psychological, social, and occupational functioning of patients and their quality of life. The presence of these conditions results in numerous short-and long-term complications and an increase in mortality compared to people with depression or diabetes alone [11].

Depression is two times more likely in people with type 1 or type 2 diabetes than in people without the disease [12]. Systematic reviews of 248 observational studies showed that 28% of adults with type 2 diabetes in the world (n = 23,245,827) had depression. Almost one in four adults with type 2 diabetes had depression. Depression prevalence was lower in Europe (24%), Africa (27%), America (28%) and higher in Australia (29%) and Asia (32%) [13].

In addition to depression and anxiety, diabetes-specific distress has shown to be prevalent among people with diabetes [14]. It is distinct from depression and refers to the fears, worries and frustrations that people experience while living with and managing diabetes [15,16].

The literature has shown that psychological disorders are present in the population with diabetes mellitus in different age groups and that there are a variety of interventions to treat these disorders in this population. However, the synthesis of these findings was not found in the literature that prompted the realization of this overview. By synthesizing the findings on effectiveness and safety of the interventions available to treat this subject, as well as highlighting the lack of information; this study can guide people with diabetes, their caregivers and health professionals in making decisions about the use of interventions in the mental health-care of these patients.

Therefore, this study summarized the available evidence regarding the effectiveness and safety of interventions to treat psychological disorders, in people with type 1 and type 2 diabetes. More specifically, the review questions are as follows: What is the effectiveness and safety of interventions to treat depression and/or anxiety in people with type 1 and type 2 diabetes? What is the quality of the evidence of these findings?.

## Methods

### Protocol and register

This review of systematic reviews was reported according to the Preferred Reporting Items for Systematic Reviews and Meta-Analysis (PRISMA) statement [17]. The study protocol was registered in PROSPERO (CRD42021224587).

### Eligibility criteria

Eligibility criteria were described using the Population, Intervention, Comparison, Outcome, and Type of study (PICOT) framework.

**Inclusion criteria.** <u>Population:</u> *children, adolescents or adults with both type 1 or type 2 diabetes mellitus and depression, depressive symptoms, and/or anxiety. Distress was also considered if the study population had depression or anxiety. We considered the diverse ways for diagnosing these diseases (clinical diagnosis or done through the use of different scales) reported in reviews.*

<u>Interventions:</u>

- non-pharmacological interventions: i) psychological (psychodynamic psychotherapy, interpersonal psychotherapy, non-directive counseling or support, among others); ii) psychoeducational (collaborative care, among others); iii) health education; iv) lifestyle interventions, among others;

- pharmacological interventions: drugs that were used in the treatment of depression and/or anxiety.

<u>Comparators:</u>

- non-pharmacological interventions: usual care or other non-pharmacological interventions;

- pharmacological interventions: active control or placebo.

<u>Outcomes:</u> effectiveness and safety outcomes described in "Measure outcomes".

<u>Type of study:</u> systematic review of randomized clinical trials followed by meta-analysis. Systematic reviews with more than one study design were included, but the collected information was restricted for those outcomes reported by randomized clinical trials.

**Exclusion criteria.** <u>Type of study:</u> systematic review in which interventions consisted only of adherence to diabetes treatment (although interventions to improve the adherence to diabetes have effects on mood, they were not designed to treat depression or anxiety. Review that contained clinical trials included in other reviews with the most recent publication date.

## Measure outcomes

**Primary outcomes.** The information was described by self-report, validated questionnaires, clinical diagnosis, or standardized interviews: depression improvement and/or remission (depression remission rate, reduction in depression severity score, and depression treatment response rate, among others); anxiety improvement and/or remission (reduced anxiety scores among others); reduction of diabetes-specific emotional distress; and quality of life improvement.

**Secondary outcomes.** The secondary outcomes were reduction in hemoglobin A1c (HbA1c) values, treatment costs, death from any cause, adverse drug reaction, adherence to treatment for diabetes, and complications arising from diabetes.

## Search methods for primary studies

**Electronic searches.** The databases searched were as follows: Cochrane Library, MEDLINE (via PubMed), EMBASE, Web of Science, and LILACS (via Virtual Health Library). The search included studies without language restrictions or time limits. We used information sources to locate the studies from the beginning to the December 1st, 2022.

## Searching other resources

The lists of references of eligible studies, reviews, and systematic reviews were checked by the reviewers to identify other possible studies. The Grey Literature Report (https://www.greylit.org/library/search) and OpenGrey (http://www.opengrey.eu/) were searched for grey literature. If necessary, the main authors of the studies were contacted for additional information.

### Search strategy

The search was conducted using the Medical Subject Headings (MeSH) terms for each disease: (diabetes mellitus) AND (depression OR depressive disorder OR anxiety) with the filter: systematic review (S4 File).

### Eligibility determination

Two reviewers (RF and IM), independently, assessed potentially relevant titles and abstracts and applied the eligibility criteria. The full texts of potentially eligible articles were obtained. These reviewers, independently, assessed the eligibility of each full text and resolved any disagreement by consensus. A third reviewer assisted with the final decision when necessary (CB). For duplicate publications, the article with the most complete data was used. The full article was requested from the author when necessary.

### Data extraction

Calibration exercises were performed for data extraction by using a standardized form of Excel. Calibration occurred by extracting of at least two studies, followed by consensus among reviewers. Data extraction was performed by the same reviewers (RF and IM) independently, with discrepancies resolved by consensus between them. The third reviewer was contacted when necessary (CB). The study authors were contacted by email, if necessary.

The data collected from the eligible studies were as follows: author and year of publication, time considered when searching for relevant studies, number of randomized clinical trials included, type of intervention (non-pharmacological or pharmacological), results of the measured outcomes, and duration of follow-up. The participant data collected were as follows: number, type of diabetes, and type of psychiatric disorder.

### Quality assessment of the studies

The Assessing the Methodological Quality of Systematic Reviews 2 (AMSTAR-2) tool was used to appraise the methodological quality of the included systematic reviews [18]. The tool evaluates the methodological aspects using 16 items and rating the confidence in the results of the review as high (no or one non-critical weakness), moderate (more than one non-critical weakness), low (one critical flaw with or without non-critical weaknesses), and critically low (more than one critical flaw with or without non-critical weaknesses). Details about the tool are described in the footnote of S2 Table.

### Data synthesis

The summary measures of the systematic reviews were described using odds ratio, relative risk and standard mean difference (SMD), followed by 95% confidence intervals (95% CI). We summarized the results using narrative synthesis according to the type of intervention and type of diabetes.

Heterogeneity was verified by the $I^2$ statistic which is classified as 0 to 25% (low heterogeneity), 50% (moderate heterogeneity), and 75% (high heterogeneity) [19].

For each outcome, the quality of evidence was collected from the systematic reviews according to the Grading of Recommendations Assessment, Development, and Evaluation (GRADE) system. In this approach, randomized clinical trials begins with high-quality evidence, but can be assessed as low-quality evidence based on one or more of the following five categories of limitations: risk of bias, inconsistency, indirect evidence, inaccuracy, and publication bias [20].

## Results

### Literature search

A total of 1,346 reviews were identified for screening. Following the removal of duplicates, review of titles and abstracts and full text evaluation, 13 systematic reviews were included (S3 File). It was not necessary to contact the study authors in order to request the full text. The list of excluded articles is presented in S2 File.

### Study characteristics

S1 Table describes the systematic reviews included according to the type of diabetes. The systematic reviews included accounted for 204 trials and 28,307 participants. Four studies included adults with type 2 diabetes and nine studies included adults with both type 1 and type 2 diabetes. No study specifically addressed the population of children and young people. All systematic reviews included adults with depression (n = 13) and four of these studies also included people with anxiety. Only two reviews reported information on the duration of diabetes. The intervention most reported was psychological (n = 09). The studies were published mainly in China and United Kingdom, between the years 2012 and 2020.

### Methodological quality of systematic reviews

In general, the studies had methodological problems, mainly because the authors did not report statements that the methods were previously established; did not explain their selection of the study designs for inclusion; did not provide list of excluded studies and did not justify the exclusions. Information on the sources of funding was reported by one study only. They also did not conduct adequate investigations of publication bias and did not discuss its likely impact on the results of the review. The systematic reviews had at least one critical flaw and then were of low quality according to AMSTAR-2 (S2 Table).

### Description of interventions and comparisons

The S1 Table describes the following interventions: psychological (n = 09); psychoeducational (n = 3); pharmacological (n = 3) and health education (n = 2). The outcomes reported by the studies were mainly about the effectiveness of the interventions: depression and anxiety improvement and/or remission, quality of life improvement, adherence to treatment for diabetes and reduction in HbA1c values. Only one study addressed the safety outcome with information on adverse drug reactions.

### Psychological interventions (n = 9 reviews) (S3 Table)

Psychological interventions are those in which problems are understood in terms of emotions, cognitions and behaviors [21] as psychodynamic psychotherapy, interpersonal psychotherapy, supportive therapy, counseling therapy, CBT, psychoanalytically informed therapies, family systems therapy, among others. They can be performed individually, in groups or among families [21–23]. CBT was the most studied intervention (five out of nine systematic reviews).

### Population specified with type 2 diabetes (n = 2)

Motivational interviewing had no effect on reduction of depressive symptoms in adults with type 2 diabetes (at 3 and 24 months of follow-up). This findings were based on clinical trials that had high heterogeneity and the quality of the evidence was not assessed [24].

One systematic review addressed intervention of social psychology, knows as psychosocial. This intervention reduced depressive and anxiety symptoms in adults with type 2 diabetes and improved glycemic values. However, the randomized clinical trials had high heterogeneity and the quality of the evidence was not assessed [22].

## Population with type 1 and type 2 diabetes (n = 7)

CBT is an organized and time-limited approach with content such as psychoeducation, behavioral activation, cognitive restructuring, and relapse prevention. It can reduce depression by identifying and evaluating negative thoughts [25,26].

CBT, CBT via telephone, and CBT via internet (websites and/or virtual platforms as sources of information) showed beneficial effects on improvement of depression and glycemic values, in adults with type 1 and type 2 diabetes, compared to usual care or a waiting list; upon medium-term follow-up (up to 6 months) in case of the intervention using CBT and CBT via telephone; and during one-month follow-up in case of CBT via internet. Meta-analysis indicated high heterogeneity between the studies, and in general, the quality of the evidence was considered low, when reported [23].

CBT showed positive effects on the response to treatment of depression in the long-term, in adults with type 1 and type 2 diabetes. There was no difference between the interventions in relation to response to anxiety, glycemic control, and the quality of life. In general, these findings were based on studies with high heterogeneity and presence of publication bias. The quality of evidence was not reported [25].

CBT improved depression scores in adults and elderly with type 1 and type 2 diabetes (12 months of follow-up), while anxiety scores and HBA1c values decreased up to 8 months post-intervention. In general, the findings varied from low to high heterogeneity, and the quality of evidence was not reported [26].

CBT improved the depression scores of adults with type 1 and type 2 diabetes compared to control group, for up to 12 months of follow-up. The findings showed low to moderate heterogeneity, and the quality of the evidence was not reported. The authors warn of some limitations in the findings, such as sample size, intervention duration, different comparators, and a reduced number of evaluated outcomes [27].

Systematic review shown the effectiveness of CBT for improving glycemic values and reducing depression scores, in adults with type 1 and type 2 diabetes. In general, the findings were highly heterogeneous and the quality of the evidence was not reported [28].

Mindfulness-based cognitive therapy (MBCT) and mindfulness-based stress reduction (MBSR) had significant effects compared to control group on reducing depression scores, reduction HbA1c values and quality of life improvement. This information is based on findings with high heterogeneity and the quality of evidence was not reported [29].

Systematic reviews provided an estimate of the effect of psychological interventions (not specified) on comorbid major depressive disorder or subthreshold depression in adults with diabetes. Reduction in the depression scores and in HbA1c values was observed. This information is based on findings with low heterogeneity, and moderate to high quality of evidence [30].

## Psychoeducational interventions (n = 3 reviews) (S4 Table)

Psychoeducational interventions provide information and guidance for diabetes care and/or psychological self-management [31]. Among this type of intervention, collaborative care is a coordinated management model of primary health care that involves doctors, nurses, mental health professionals, and other professionals who provide patient-oriented management based on guidelines at this level of care [31,32].

### Population specified with type 2 diabetes (n = 1)

One systematic review described the results of psychoeducational interventions grouped (did not show results by type of intervention). The interventions reduced diabetes-specific emotional distress and HbA1c values, in adults with type 2 diabetes, compared to controls (in general, defined as usual care). Most of the meta-analyses that explored the effect of specific interventions (subgroup analysis) verified no significant results supporting each intervention. However, a reduction in diabetes-specific emotional distress levels was observed for collaborative care compared to usual care. A reduction in HbA1c values for "digital platform" intervention was also compared to usual care. Such findings need to be confirmed since the quality of the evidence is low [31].

### Collaborative care intervention (n = 2 reviews) (S4 Table)

*Population with type 1 and type 2 diabetes (n = 2)*. Systematic review showed positive effects of collaborative care on the rate of response to depression treatment in adults at 6, 12 and 24 months of follow-up. However, it did not seem to benefit in long-term depression remission (12 and 24 months). The intervention was also significantly associated with higher rates of adherence to antidepressant medication and oral hypoglycemic agents. Meta-analyses showed that improvement in depression outcomes were not accompanied by significant differences in HbA1c values between the groups, until 24 months of follow-up. The quality of evidence was not reported [32].

Other systematic review observed a moderate effect on depression score reduction and improvement of glycemic values with the use of collaborative care. The quality of evidence was moderate [30].

### Health education (n = 2 reviews) (S4 Table)

**Population specified with type 2 diabetes (n = 1).** Web-based health education (via telephone calls or SMS—short message service texts) provided information on education, peer support, and/or overall therapeutic components for adults with type 2 diabetes. The results did not differ in relation to depression and anxiety scores compared to control groups (usual care or sessions with a health professional or minimal computer support, among others). High heterogeneity was observed among studies and the quality of the evidence was not assessed [33].

**Population with type 1 and type 2 diabetes (n = 1).** Health education was superior to usual care in the depression remission and glycemic control. However, this information is based on findings with medium to high heterogeneity and moderate quality of evidence [23].

### Pharmacological interventions (n = 3 reviews) (S5 Table)

**Population with type 1 and type 2 diabetes (n = 3).** Specific serotonin reuptake inhibitors (SSRIs) showed improvement in short-term depression severity scores, short-term depression remission and glycemic control, in adults with type 1 and type 2 diabetes, compared to placebo. However, the evidence was of low quality, and the patients' follow-up period did not exceed six months. The authors reported that other outcomes, such as medium- and long-term depression and glycemic control outcomes, as well as healthcare costs, diabetes complications and mortality, were not evaluated in clinical trials. The review did not collect information on the adverse drug reaction [23].

Pharmacological treatment (mainly SSRIs) showed significant effects in terms of depression outcomes, but small in terms of glycemic values. Although interventions were effective for depression, they were not effective for glycemic control. However, the quality of evidence was

not performed by review. The review did not collect information on the adverse drug reaction [30].

*Gardenia Fructus* antidepressant formula significantly improved the depressive symptoms in people compared to no antidepressant treatment. However, the formula increased the incidence of headache/dizziness and diarrhea. *G. Fructus* antidepressant formula alone or in combination with SSRIs, reduced the depressive symptoms compared to SSRIs alone. Sensitivity analysis suggests that the findings could be inconclusive due to study heterogeneity (in terms of diagnostic criteria or methodological quality). The incidence of adverse events in the *G. Fructus* antidepressant formula group and the herbal medicine plus SSRI group was not statistically different from SSRI group [34].

## Discussion

### Summary of the main findings

This study included 13 systematic reviews, nine of which included people with both type 1 and type 2 diabetes and four included population with specific type 2 diabetes. No study reported only children and adolescents. The population described was mainly of adults with type 1 or type 2 diabetes and with depression.

In general, the interventions seem be effectiveness for most of the evaluated outcomes and mainly for depression, since only four reviews reported population with anxiety. The safety outcomes were restricted to report of adverse drug reactions, reported in only one review. It is important to emphasize that the reviews had methodological problems, which may limit confidence in the findings.

CBT is an organized and time-limited approach with content such as psychoeducation, behavioral activation, cognitive restructuring, and relapse prevention. It can reduce depression by identifying and evaluating negative thoughts. It can be useful for people with chronic illness by way of improving self-care skills and learning to adjust to the disease and its impact on their daily lives [25,26]. This intervention showed promising results for most of the evaluated outcomes and mainly for depression, in adults with type 1 or type 2 diabetes. The anxiety outcome was reported in two studies, with divergent results between them [25,26]. The divergences can be explained by differences in clinical trials included in reviews, since there was an overlap of around 30% of these studies included in both reviews. One systematic review observed improvement of the quality of life comparing CBT with usual care [23]. In general, the findings were heterogeneous and there was lack of detail of reported evidence by three of five systematic reviews.

Other psychological interventions evaluated were psychosocial, mindfulness-based cognitive therapy, mindfulness-based stress reduction and motivational interviewing; except for the latter, the other interventions reduced depressive symptoms in adults.

Among psychoeducational interventions, two systematic reviews about collaborative care shown that the depression treatment response rate/depression scores, the HBA1c values and the adherence to treatment had improvements with this intervention. These findings were based on low to moderate quality of evidence [30,32]. The use of a digital platform reduced HbA1c values compared to usual care [31].

Health education was superior to usual care in the depression remission and glycemic control, in adults with type 1 and type 2 diabetes (moderate quality of evidence) [23]. Other review did not observe benefits of this web-based intervention to reduce depression or distress rate, in adults with type 2 diabetes [33].

Three systematic reviews addressed the use of drugs to treat depression in people with diabetes. The results shown that SSRIs seem to improve short-term depression severity scores,

short-term depression remission and glycemic control compared with placebo [23]. Other review show that the use of herbal medicine (*Gardenia Fructus*) alone and combined with SSRIs, improved the depressive symptoms compared to placebo and SSRI alone [34]. The authors of reviews concluded that the results should be confirmed in people with type 1 and type 2 diabetes. It is important to highlight that some precautions must be considered in relation to the use of antidepressants, since that these drugs can interfere with the weight gain and the metabolic changes in the population with diabetes [35].

## Strengths and limitations of the study

This review summarized the available evidence on interventions to treat depression and/or anxiety in people with type 1 and type 2 diabetes. Although the quality of the evidence reported was restricted to the information described by systematic reviews and can be affected by the methodological quality of these studies; the benefit was to broadly demonstrate what the literature reports about the subject, as well as to highlight the lack of information. Furthermore, this study included a comprehensive literature search, all stages of selection and data extraction were performed, in both pairs and independently, and there was no language restriction with respect to the reviews included.

Of a total of 204 clinical trials identified within the systematic reviews, 37.8% of them were included in more than one review. There was no overlap of clinical trials about to the intervention's health education, collaborative care and some psychological interventions. Of the 66 clinical trials (included in the five systematic reviews) about CBT intervention, 31.0% of them had overlapped. Two reviews evaluating pharmacological interventions presented overlap in 50% of the articles included, however, one of them described in more detail the drugs and the evaluated outcomes.

It is important to emphasize that safety outcomes were restricted to report of adverse drug reactions and quality of life was assessed in only two systematic reviews. Costs, diabetes complications and mortality were not reported by clinical trials, although they are relevant outcomes to assessing the effectiveness and safety of interventions health care [23,31]. The reviews too considered include clinical trials with different scales used for diagnosis of psychological disorders.

Most of the systematic reviews included adults with type 1 and type 2 diabetes and did not perform subgroup analyses for the type of psychological diseases (depression or anxiety), age or follow-up time. These findings could explain the heterogeneity observed in results, since psychological diseases can manifest themselves differently in people with type 1 and type 2 diabetes and the follow-up time also differed between the clinical trials.

## Implications for clinical practice and research

CBT, collaborative care, health education (except web-based health education) and pharmacological interventions showed positive results, mainly for the treatment of depression and improve glycemic values of people with diabetes.

CBT showed to have promising results for most of the evaluated outcomes. This intervention showed to be more effective when the frequency of the meetings was at least four times a month [23]. Collaborative care can also be an effective tool for professionals who want to improve medication adherence in these people [32]. Health education can promote well-being in people with diabetes and emotional diabetes management [33]. The pharmacological intervention was evaluated in few studies and this could have occurred, since trials on the treatment of depression are not limited to a specific population.

Although the interventions were effective in most of the evaluated outcomes, it is important to emphasize that the reviews had methodological problems, as well as there was heterogeneity

in reported outcomes and absence of report of some outcomes. The quality of the evidence was not cited in some reviews and, when evaluated, varied from low to moderate. It was observed that no systematic review about the population with anxiety assessed the quality of the evidence, limiting confidence in its findings. The population of children and adolescents has been little studied. Too no review addressed the effectiveness of combined interventions, which can be adopted to treat depression and/or anxiety in this population. In the face of these reports, additional clinical trials can confirm our findings and expand the scope of the research to include groups of people with specific age, types of diabetes and psychological disorders (depression, anxiety or diabetes-specific distress).

## Conclusion

The interventions reported in literature can be effective to treat mainly adults with diabetes and depression and CBT showed promising results for treat this population for most of the evaluated outcomes. Considering the limitations observed in the reported of quality of evidence, future randomized clinical trials can confirm these findings, as well as consider the gaps identified in the literature. This study can guide patients, their caregivers and health professionals in making decisions concerning the use of these interventions in the mental healthcare of people with diabetes.

## Supporting information

**S1 Table. Characteristics of systematic reviews included (n = 13).**
(DOCX)

**S2 Table. Methodological quality of the systematic reviews' assessment by AMSTAR-2 (n = 13).**
(DOCX)

**S3 Table. Results of psychological interventions (n = 9 reviews).**
(DOCX)

**S4 Table. Results of psychoeducational interventions (n = 3 reviews) and health education (n = 2 reviews).**
(DOCX)

**S5 Table. Results of pharmacological interventions (n = 3 reviews).**
(DOCX)

**S1 File. Data information file.**
(XLSX)

**S2 File. Characteristics of excluded studies.**
(DOCX)

**S3 File. Flow diagram.**
(DOCX)

**S4 File. Search strategy.**
(DOCX)

## Author Contributions

**Conceptualization:** Reginaldo Tavares Franquez, Cristiane de Cássia Bergamaschi.

**Data curation:** Reginaldo Tavares Franquez, Cristiane de Cássia Bergamaschi.

**Formal analysis:** Reginaldo Tavares Franquez, Isabela Muniz de Souza, Cristiane de Cássia Bergamaschi.

**Funding acquisition:** Cristiane de Cássia Bergamaschi.

**Investigation:** Reginaldo Tavares Franquez, Cristiane de Cássia Bergamaschi.

**Methodology:** Reginaldo Tavares Franquez, Isabela Muniz de Souza, Cristiane de Cássia Bergamaschi.

**Project administration:** Cristiane de Cássia Bergamaschi.

**Supervision:** Cristiane de Cássia Bergamaschi.

**Validation:** Reginaldo Tavares Franquez, Cristiane de Cássia Bergamaschi.

**Visualization:** Reginaldo Tavares Franquez, Isabela Muniz de Souza, Cristiane de Cássia Bergamaschi.

**Writing – original draft:** Reginaldo Tavares Franquez, Isabela Muniz de Souza, Cristiane de Cássia Bergamaschi.

**Writing – review & editing:** Reginaldo Tavares Franquez, Isabela Muniz de Souza, Cristiane de Cássia Bergamaschi.

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
