## [Decision Letter · Decision Letter 0]

3 Aug 2022

PONE-D-22-05027Interventions for depression and anxiety among people with diabetes mellitus: an overview of systematic reviews and meta-analysesPLOS ONE

Dear Dr. Bergamaschi,

Thank you for submitting your manuscript to PLOS ONE. After careful consideration, we feel that it has merit but does not fully meet PLOS ONE’s publication criteria as it currently stands. Therefore, we invite you to submit a revised version of the manuscript that addresses the points raised during the review process.

Please note that we have only been able to secure a single reviewer to assess your manuscript. We are issuing a decision on your manuscript at this point to prevent further delays in the evaluation of your manuscript. Please be aware that the editor who handles your revised manuscript might find it necessary to invite additional reviewers to assess this work once the revised manuscript is submitted. However, we will aim to proceed on the basis of this single review if possible.  The reviewer raised a number of concerns that need attention. They request additional information on methodological aspects of the study such as finer details report age range, inclusion criteria of individual reviews, and diabetes duration. Could you please revise the manuscript to carefully address the concerns raised?  Please submit your revised manuscript by Sep 16 2022 11:59PM. If you will need more time than this to complete your revisions, please reply to this message or contact the journal office at plosone@plos.org. Please include the following items when submitting your revised manuscript:A rebuttal letter that responds to each point raised by the academic editor and reviewer(s). You should upload this letter as a separate file labeled 'Response to Reviewers'.A marked-up copy of your manuscript that highlights changes made to the original version. You should upload this as a separate file labeled 'Revised Manuscript with Track Changes'.An unmarked version of your revised paper without tracked changes. You should upload this as a separate file labeled 'Manuscript'.If applicable, we recommend that you deposit your laboratory protocols in protocols.io to enhance the reproducibility of your results. Protocols.io assigns your protocol its own identifier (DOI) so that it can be cited independently in the future. For instructions see: https://journals.plos.org/plosone/s/submission-guidelines#loc-laboratory-protocols. Additionally, PLOS ONE offers an option for publishing peer-reviewed Lab Protocol articles, which describe protocols hosted on protocols.io. Read more information on sharing protocols at https://plos.org/protocols?utm_medium=editorial-email&utm_source=authorletters&utm_campaign=protocols.

We look forward to receiving your revised manuscript.

Kind regards,

Katrien Janin, PhD

Staff Editor

PLOS ONE

Journal Requirements:

Reviewers' comments:

Reviewer's Responses to Questions

**Comments to the Author**

1. Is the manuscript technically sound, and do the data support the conclusions?

Reviewer #1: Yes

2. Has the statistical analysis been performed appropriately and rigorously? 

Reviewer #1: N/A

3. Have the authors made all data underlying the findings in their manuscript fully available?

Reviewer #1: Yes

4. Is the manuscript presented in an intelligible fashion and written in standard English?

Reviewer #1: Yes

5. Review Comments to the Author

Reviewer #1: This is an important review of reviews of interventions for depression and anxiety for people with diabetes. This area is well researched so it’s important to bring together all the literature in one place.

I would expect something like this to be called ‘a review of reviews’

Is it a review of meta-analyses if you are narratively synthesising the information?

Might be appropriate to use the vote count method (which would be straightforward based on what you have achieved already), see SWiM guidelines: https://www.bmj.com/content/368/bmj.l6890

Abstract

Clarify ‘standard interventions’- do you mean usual care?

‘Among others’ is too vague- please specify. If short on words, only report primary outcomes in abstract.

‘Reviewers, in pairs and independently’ is unclear.

Please expand on ‘narrative synthesis’- did you tabulate results, organise by any group (I assume intervention type?)?

Introduction

Diabetes is not a ‘disease’, it is a long-term health condition.

Avoid terms such as ‘suffering’ and ‘control’ and ‘patients’, see language matters guidance: https://www.england.nhs.uk/wp-content/uploads/2018/06/language-matters.pdf

Avoiding multiple prevalence data, with different stats representing same information (i.e. in first paragraph).

Often unclear whether data and studies involve T1D and/or T2D. I would expect to see stronger justification for including both T1D and T2D, and not just focusing on one type.

You allude to safety in your aims, but this isn’t introduced in the background research- please provide rational for this inclusion.

‘Regardless of their age’ is an odd phrase. Are you looking to compare adolescent’s vs adults? Consider phraseology here. It looks like you go on to only review studies with adults- so you might not need this clause in the research aim/question.

Methods

Ethics statement not need for SR.

Misplaced bracket in line 123.

Outcomes should be more specific in eligibility criteria.

Could consider adding a table to represent inclusion/exclusion criteria.

Again, clarify ‘Pairs of reviewers (RF and IM) and independently’

How many articles were requested from authors? How many did you obtain from this process? (this might belong in results section)

Results

The study characteristics section seems quite limited. I would expect to see a PCIOs break down as reported in methods. E.g. Can you finer details report age range, inclusion criteria of individual reviews, diabetes duration etc.

It would be useful to see comment on the measures used for depression and anxiety across reviews.

6. PLOS authors have the option to publish the peer review history of their article (what does this mean?). If published, this will include your full peer review and any attached files.

Reviewer #1: No

---

## [Author Response · Author response to Decision Letter 0]

6 Sep 2022

Reviewers' comments: 

1. Is the manuscript technically sound, and do the data support the conclusions?

Reviewer #1: Yes

2. Has the statistical analysis been performed appropriately and rigorously?

Reviewer #1: N/A

3. Have the authors made all data underlying the findings in their manuscript fully available?

Reviewer #1: Yes

4. Is the manuscript presented in an intelligible fashion and written in standard English? PLOS ONE does not copyedit accepted manuscripts, so the language in submitted articles must be clear, correct, and unambiguous. Any typographical or grammatical errors should be corrected at revision, so please note any specific errors here.

Reviewer #1: Yes

Dear Reviewer 

Thank you for giving us another opportunity to answer the comments made by reviewer in our manuscript. We answer all comments and remain at your disposal if further adjustments are necessary. 

1. This is an important review of reviews of interventions for depression and anxiety for people with diabetes. This area is well researched so it’s important to bring together all the literature in one place. I would expect something like this to be called ‘a review of reviews. Is it a review of meta-analyses if you are narratively synthesising the information? Might be appropriate to use the vote count method (which would be straightforward based on what you have achieved already), see SWiM guidelines: https://www.bmj.com/content/368/bmj.l6890

Answer: Dear reviewer, thank you for the suggestions that contributed to the improvement of the manuscript. We agree with the change in the writing of the study design “overview of systematic reviews and meta-analyses”. We carried out a search on Prospero and the most of the protocols cite this type of study as: “review of systematic reviews”. Then, we modified the title, but if the reviewer deems another adjustment necessary, we remain at your disposal. 

2. Abstract

Clarify ‘standard interventions’- do you mean usual care? ‘Among others’ is too vague- please specify. If short on words, only report primary outcomes in abstract. ‘Reviewers, in pairs and independently’ is unclear. Please expand on ‘narrative synthesis’- did you tabulate results, organize by any group (I assume intervention type?)?

Answer: Dear reviewer, we accept all suggestions and we have rewritten the parts of the abstract in which changes were requested.

3. Introduction

Diabetes is not a ‘disease’, it is a long-term health condition. Avoid terms such as ‘suffering’ and ‘control’ and ‘patients’, see language matters guidance: https://www.england.nhs.uk/wp-content/uploads/2018/06/language-matters.pdf

Answer: We accept the suggestions and adapt most of the text with the suggested words. We have changed the term “glycemic control” to “glycemic values” in most of the text. “Glycemic control” was a term well used by the included reviews and sometimes it was necessary to keep it. However, we remain at your disposal if any other adjustments are necessary.

4. Avoiding multiple prevalence data, with different stats representing same information (i.e. in first paragraph). Regardless of their age’ is an odd phrase. Are you looking to compare adolescent’s vs adults? Consider phraseology here. It looks like you go on to only review studies with adults- so you might not need this clause in the research aim/question.

Answer: We have made the adjustments as requested.

5. Often unclear whether data and studies involve T1D and/or T2D. I would expect to see stronger justification for including both T1D and T2D, and not just focusing on one type. You allude to safety in your aims, but this isn’t introduced in the background research- please provide rational for this inclusion.

Answer: We included studies with both type 1 and type 2 diabetes since the literature have demonstrated that psychological disorders are present in individuals with both diseases. When we read the full text of the included reviews, most of the reviews included both the type 1 and type 2 diabetes population, as well as the meta-analyses pooled information from both populations. Of the 13 reviews, 9 included both populations. We sought to make this information clearer in the “Introduction” and “Discussion” of the study.

Little information is available in the literature regarding the safety of interventions. In our results, this information was restricted to report of adverse reactions to antidepressants. Since safety was a secondary outcome in our study, we chose not to highlight it in the introduction, but we inserted this information in “Results” and pointed out this limitation of the studies in “Discussion”.

6. Methods

Ethics statement not need for SR. Misplaced bracket in line 123. Again, clarify ‘Pairs of reviewers (RF and IM) and independently’

Answer: We have made the adjustments as requested.

7. Outcomes should be more specific in eligibility criteria. Could consider adding a table to represent inclusion/exclusion criteria.

Answer: We have inserted more detailed information on outcomes in “Outcomes evaluated”, since the table would be disproportionate with more information on inclusion than exclusion criteria. However, we remain at your disposal if further adjustments are necessary.

8. How many articles were requested from authors? How many did you obtain from this process? (This might belong in results section)

Answer: We enter this information in the “Results” as requested.

9. Results

The study characteristics section seems quite limited. I would expect to see a PCIOs break down as reported in methods. E.g. Can you finer details report age range, inclusion criteria of individual reviews, diabetes duration etc. It would be useful to see comment on the measures used for depression and anxiety across reviews.

Answer: We made some adjustments in “Results”. We placed the “psychiatric illness” column before “intervention” column to bring it closer to population information. We highlighted the duration of diabetes disease in the footnote, as only two reviews provided this information. The reviews used different tools and scales to measure anxiety and depression and this information is available in S3 Appendix (Supporting information file).

---

## [Decision Letter · Decision Letter 1]

21 Nov 2022

PONE-D-22-05027R1Interventions for depression and anxiety among people with diabetes mellitus: review of systematic reviewsPLOS ONE

Dear Dr. Bergamaschi,

Thank you for submitting your manuscript to PLOS ONE. After careful consideration, we feel that it has merit but does not fully meet PLOS ONE’s publication criteria as it currently stands. Therefore, we invite you to submit a revised version of the manuscript that addresses the points raised during the review process.

Your manuscript has been assessed by an additional reviewer whose report can be found below. As you will see from the comments, there remain some concerns which should be addressed before your manuscript is suitable for publication. In particular, we request that you update your search to include the most recent studies. 

We look forward to receiving your revised manuscript.

Kind regards,

Dr Joseph Donlan

Senior Editor

PLOS ONE

Reviewers' comments:

Reviewer's Responses to Questions

**Comments to the Author**

1. If the authors have adequately addressed your comments raised in a previous round of review and you feel that this manuscript is now acceptable for publication, you may indicate that here to bypass the “Comments to the Author” section, enter your conflict of interest statement in the “Confidential to Editor” section, and submit your "Accept" recommendation.

Reviewer #2: (No Response)

2. Is the manuscript technically sound, and do the data support the conclusions?

Reviewer #2: Partly

3. Has the statistical analysis been performed appropriately and rigorously? 

Reviewer #2: (No Response)

4. Have the authors made all data underlying the findings in their manuscript fully available?

Reviewer #2: No

5. Is the manuscript presented in an intelligible fashion and written in standard English?

Reviewer #2: Yes

6. Review Comments to the Author

Reviewer #2: Dear Authors,

Unfortunately, there are several fundamental errors.

1- The last search is July 2021, which means you lost several related studies during more than one year ago. On the other hand, it is currently published some meta-analyses in the topic:

----Mersha AG, Tollosa DN, Bagade T, Eftekhari P. A bidirectional relationship between diabetes mellitus and anxiety: A systematic review and meta-analysis. J Psychosom Res. 2022 Nov;162:110991. doi: 10.1016/j.jpsychores.2022.110991.

----Association between social supports and depression among patients with diabetes mellitus in Ethiopia: a systematic review and meta-analysis. BMJ Open. 2022 May 11;12(5):e061801.

2-At the first should be clear definition of depression, and anxiety, and then address PICO.

3- There are some concerns in PICO. It is said "psychoeducational interventions (including information and guidance for diabetes and/or psychological self-management" as intervention group and "non-pharmacological interventions may include

usual care or other interventions" as Comparators. In your mention "information and guidance....." are pharmacological or non-pharmacological interventions? Considering that they are an example of non-pharmacological interventions, why is chosen

"non-pharmacological interventions as comparison group"?

4- What are exclusion criteria?

5- The data on world prevalence of diabetes is old which should be updated.

6-One of the limitation of the study is heterogeneity in anxiety assessment scales.

7-In search strategy, there are similarity between #3 and #4 and also between #5 and #6, which was better to delete common words before beginning search.

Best Regards,

7. PLOS authors have the option to publish the peer review history of their article (what does this mean?). If published, this will include your full peer review and any attached files.

Reviewer #2: No

---

## [Author Response · Author response to Decision Letter 1]

16 Dec 2022

December, 14nd 2022

Dear editor Joseph Donlan 

Thank you for giving us the opportunity to answer the comments made by reviewer in our manuscript entitled: “Interventions for depression and anxiety among people with diabetes mellitus: an overview of systematic reviews and meta-analyses" (PONE-D-22-05027). 

We have inserted all the information requested by the reviewer and updated the bibliographic search in the databases carried out in December 2022. The authors look forward to a response as soon as possible.

We are looking forward to hearing from you. 

 Sincerely,

Dr. Cristiane de Cássia Bergamaschi

University of Sorocaba - Sorocaba/SP, Brazil

Reviewers' comments: 

1. If the authors have adequately addressed your comments raised in a previous round of review and you feel that this manuscript is now acceptable for publication, you may indicate that here to bypass the “Comments to the Author” section, enter your conflict-of-interest statement in the “Confidential to Editor” section, and submit your "Accept" recommendation.

Reviewer #2: (No Response)

2. Is the manuscript technically sound, and do the data support the conclusions?

The manuscript must describe a technically sound piece of scientific research with data that supports the conclusions. Experiments must have been conducted rigorously, with appropriate controls, replication, and sample sizes. The conclusions must be drawn appropriately based on the data presented. Reviewer #2: Partly

Answer: Dear reviewer, thank you for the suggestions that contributed to the improvement of the manuscript. We redid the search strategy and identified 10 systematic reviews with potential to be included in the study. After reading of the full texts, none of them met the eligibility criteria for our study. These reviews were included in the list of excluded studies (S2 Appendix) and the references are highlighted “in blue”. We remain at your disposal if any other adjustment will be necessary.

3. Has the statistical analysis been performed appropriately and rigorously?

Reviewer #2: (No Response)

4. Have the authors made all data underlying the findings in their manuscript fully available?

Reviewer #2: No

Answer: All the study data is available in an excel spreadsheet. This file is named as “S3 Appendix”.

5. Is the manuscript presented in an intelligible fashion and written in standard English?

PLOS ONE does not copyedit accepted manuscripts, so the language in submitted articles must be clear, correct, and unambiguous. Any typographical or grammatical errors should be corrected at revision, so please note any specific errors here. Reviewer #2: Yes

6. Review Comments to the Author: Dear Authors, unfortunately, there are several fundamental errors.

1- The last search is July 2021, which means you lost several related studies during more than one year ago. On the other hand, it is currently published some meta-analyses in the topic: Mersha AG, Tollosa DN, Bagade T, Eftekhari P. A bidirectional relationship between diabetes mellitus and anxiety: A systematic review and meta-analysis. J Psychosom Res. 2022 Nov;162:110991. doi: 10.1016/j.jpsychores.2022.110991. Association between social supports and depression among patients with diabetes mellitus in Ethiopia: a systematic review and meta-analysis. BMJ Open. 2022 May 11;12(5):e061801.

Answer: As requested by the reviewer, we redid the search strategy and identified 10 potentially eligible systematic reviews. After selecting the studies based on reading the full texts, none of them met the eligibility criteria. The S2 Appendix contain the list of excluded studies along with the justifications and information highlighted in blue. The studies cited by the reviewer were not included in “results”, since they are reviews that reported disease prevalence data. We inserted one of reviews in “introduction”.

2- At the first should be clear definition of depression, and anxiety, and then address PICO.

Answer: The systematic reviews included reported different definitions and considered different scales for diagnosing depression, depressive symptoms and anxiety in the studied population. This information is available in the S3 Appendix. We have modified the text of method to insert this information. We appreciated the suggestion and we are available to make to make any changes that the reviewer deems necessary.

“Population: children, adolescents or adults with both type 1 or type 2 diabetes mellitus and with depression, depressive symptoms, and/or anxiety. Distress was also considered if the study population had depression or anxiety. We considered the diverse ways for diagnosing these diseases (clinical diagnosis or by using of different scales) reported in systematic reviews”.

3- There are some concerns in PICO. It is said "psychoeducational interventions (including information and guidance for diabetes and/or psychological self-management" as intervention group and "non-pharmacological interventions may include

usual care or other interventions" as Comparators. In your mention "information and guidance" are pharmacological or non-pharmacological interventions? Considering that they are an example of non-pharmacological interventions, why is chosen

"non-pharmacological interventions as comparison group"? 4- What are exclusion criteria?

Answer: We've modified the text “Eligibility criteria” to make the information clearer. Studies could compare different non-pharmacological interventions. Thus, we found “usual care” or another non-pharmacological intervention as comparator group.

Eligibility criteria 

Inclusion criteria

Interventions: 

non-pharmacological interventions: i) psychological (psychodynamic psychotherapy, interpersonal psychotherapy, non-directive counseling or support, among others); ii) psychoeducational (collaborative care, among others); iii) health education; iv) lifestyle interventions, among others; 

pharmacological intervention: drugs used in the treatment of depression and/or anxiety.

Comparators: 

non-pharmacological interventions: usual care or other non-pharmacological interventions;

pharmacological interventions: active control or placebo.

Outcomes: effectiveness and safety outcomes described in “Measure outcomes”.

Type of study: systematic review of randomized clinical trials followed by meta-analysis. Systematic reviews with more than one study design were included, but the collected information was restricted for those outcomes reported by randomized clinical trials.

Exclusion criteria

Type of study: systematic review in which interventions consisted only of adherence to diabetes treatment (although interventions to improve the adherence to diabetes have effects on mood, they were not designed to treat depression or anxiety. Review that contained clinical trials included in other reviews with the most recent publication date.

5- The data on world prevalence of diabetes is old which should be updated.

Answer: Dear reviewer, we have updated the information as requested in “introduction”.

6- One of the limitations of the study is heterogeneity in anxiety assessment scales.

Answer: Dear reviewer, we have inserted this information in discussion “Study limitations”.

7-In search strategy, there are similarity between #3 and #4 and also between #5 and #6, which was better to delete common words before beginning search.

Answer: In “S1 Appendix. “Search strategy” we insert the Mesh terms used in the search performed in MEDLINE and adapted for the other databases. We inserted the terms without repetition, however, the database MEDLINE the file that we make available in Appendix S1.

Dear reviewer, we appreciated the suggestion and are at your disposal for any adjustments that the reviewer deems necessary.

---

## [Decision Letter · Decision Letter 2]

23 Jan 2023

Interventions for depression and anxiety among people with diabetes mellitus: review of systematic reviews

PONE-D-22-05027R2

Dear Dr. Bergamaschi,

We’re pleased to inform you that your manuscript has been judged scientifically suitable for publication and will be formally accepted for publication once it meets all outstanding technical requirements.

Kind regards,

Ying Lau, Ph.D

Academic Editor

PLOS ONE

Additional Editor Comments (optional):

Dear Authors.

Thank you very much for addressing comments of reviewers.

Regards,

Lau Ying

Reviewers' comments:

Reviewer's Responses to Questions

**Comments to the Author**

1. If the authors have adequately addressed your comments raised in a previous round of review and you feel that this manuscript is now acceptable for publication, you may indicate that here to bypass the “Comments to the Author” section, enter your conflict of interest statement in the “Confidential to Editor” section, and submit your "Accept" recommendation.

Reviewer #2: (No Response)

Reviewer #3: (No Response)

2. Is the manuscript technically sound, and do the data support the conclusions?

Reviewer #2: (No Response)

Reviewer #3: Yes

3. Has the statistical analysis been performed appropriately and rigorously? 

Reviewer #2: (No Response)

Reviewer #3: Yes

4. Have the authors made all data underlying the findings in their manuscript fully available?

Reviewer #2: (No Response)

Reviewer #3: Yes

5. Is the manuscript presented in an intelligible fashion and written in standard English?

Reviewer #2: (No Response)

Reviewer #3: Yes

6. Review Comments to the Author

Reviewer #2: Dear Authors,

Although the quality of the manuscript is significantly improved, it is suggested considering language editing. For example:

-Please change "reviews" to "review studies" in line 220 and in other places.

-Re-check whole manuscript and edit such as "The reviews too considered include clinical trials with different scales used for diagnosis of psychological disorders" line 495.

-Please remove this sentence "It was not necessary to contact the study authors in order to request the full text." from line 223.

-Ref. 2 is incorrect. You addressed to recently published data of diabetes prevalence but addressed it published on 2013?

Sincerely,

Reviewer #3: I have found your paper very significant. It deals with the interventions for depression and anxiety among people with diabetes. The theory of the paper sounds logical enough. The methodology was appropriately designed, following the necessary steps (AMSTAR-2). Results were presented clearly. Discussion was detailed and clear. Moreover, the use of English is excellent, as well as style and format of the whole manuscript. The study received a scholarship, which is mentioned. In my point of view, it is an excellent paper which contributes immensely to the people with diabetes.

7. PLOS authors have the option to publish the peer review history of their article (what does this mean?). If published, this will include your full peer review and any attached files.

Reviewer #2: No

Reviewer #3: No

---

## [Editor Report · Acceptance letter]

31 Jan 2023

PONE-D-22-05027R2 

Interventions for depression and anxiety among people with diabetes mellitus: review of systematic reviews 

Dear Dr. Bergamaschi:

I'm pleased to inform you that your manuscript has been deemed suitable for publication in PLOS ONE. Congratulations! Your manuscript is now with our production department. 

Kind regards, 

on behalf of

Dr. Ying Lau 

Academic Editor

PLOS ONE